# Persistence of Schistosomiasis-Related Morbidity in Northeast Brazil: An Integrated Spatio-Temporal Analysis

**DOI:** 10.3390/tropicalmed6040193

**Published:** 2021-10-28

**Authors:** Bárbara Morgana da Silva, Anderson Fuentes Ferreira, José Alexandre Menezes da Silva, Rebeca Gomes de Amorim, Ana Lúcia Coutinho Domingues, Marta Cristhiany Cunha Pinheiro, Fernando Schemelzer de Moraes Bezerra, Jorg Heukelbach, Alberto Novaes Ramos

**Affiliations:** 1Postgraduate Programme in Public Health, School of Medicine, Federal University of Ceará, Fortaleza 60430-140, Brazil; anderson_deco.f2@hotmail.com; 2NHR Brasil—Nederlanse Stichting Voor Leprabestrijding, Fortaleza 60170-001, Brazil; alexandre@nhrbrasil.org.br (J.A.M.d.S.); malikbeca@hotmail.com (R.G.d.A.); 3Department of Nursing, Federal University of Ceará, Fortaleza 60430-160, Brazil; 4Graduate Program in Tropical Medicine, Centre for Health Sciences, Federal University of Pernambuco, Recife 50670-901, Brazil; alcdomingues@hotmail.com; 5Clinical Hospital, Federal University of Pernambuco, Recife 50670-901, Brazil; 6Laboratory of Research in Molluscan Parasitology and Biology, Department of Clinical and Toxicological Analysis, Federal University of Ceará, Fortaleza 60430-160, Brazil; marta.pinheiro@ufc.br (M.C.C.P.); bezerra@ufc.br (F.S.d.M.B.); 7Department of Community Health, School of Medicine, Federal University of Ceará, Fortaleza 60430-235, Brazil

**Keywords:** schistosomiasis mansoni, epidemiology, public health surveillance, morbidity

## Abstract

Objective: To analyze the temporal trend and spatial patterns of schistosomiasis-related morbidity in Northeast Brazil, 2001–2017. Methods: Ecological study, of time series and spatial analysis, based on case notifications and hospital admission data, as provided by the Ministry of Health. Results: Of a total of 15,574,392 parasitological stool examinations, 941,961 (6.0%) were positive, mainly on the coastline of Pernambuco, Alagoas and Sergipe states. There was a reduction from 7.4% (2002) to 3.9% (2017) of positive samples and in the temporal trend of the detection rate (APC—11.6*; Confidence Interval 95%—13.9 to −9.1). There was a total of 5879 hospital admissions, with 40.4% in Pernambuco state. The hospitalization rate reduced from 0.82 (2001) to 0.02 (2017) per 100,000 inhabitants. Conclusion: Despite the reduction in case detection and hospitalizations, the persistence of focal areas of the disease in coastal areas is recognized. This reduction may indicate a possible positive impact of control on epidemiological patterns, but also operational issues related to access to healthcare and the development of surveillance and control actions in the Unified Health System.

## 1. Introduction

Intestinal schistosomiasis caused by the trematode *Schistosoma mansoni* is a neglected tropical disease (NTD) of chronic evolution, strongly associated with the absence of basic sanitation [1]. It is a persistent public health problem, considering the associated high morbidity and mortality burden in different countries in South America, particularly in Brazil [2]. The clinical disease may vary from asymptomatic to more severe clinical forms, which may lead to death [3].

The new World Health Organization (WHO) guidelines for the control of NTDs set a global goal of eliminating the disease as a public health problem by 2030 [4]. A total of 78 countries are expected to achieve disease elimination by 2030, defined as <1% high-intensity schistosomiasis infections [4,5].

In Brazil, control actions began with the Special Programme for the Control of Schistosomiasis (Programa Especial de Controle da Esquistossomose: PECE) in the 1970s, which enabled the development of systematic actions based on the national plan that recommended the use of preventive chemotherapy and the application of molluscicides in water [6]. The aim was to control the transmission of the disease and, above all, to reduce the estimated prevalence of cases, as well as the occurrence of deaths in endemic areas [6]. However, the disease has maintained its endemic character and remains responsible for a significant burden of morbidity and mortality in the country [2,7,8]. It is estimated that approximately 1.5 million people live in high-risk areas risk of transmission, with Brazil’s southeast and northeast regions being most affected [9].

Biological, demographic, cultural, political, and socioeconomic factors have contributed to the maintenance of endemicity [10]. Despite its predominantly rural origin, since the 1990s, there has been a significant number of cases in urban areas, especially in coastal areas of the northeast region [11]. In this Brazilian region, transmission is well established in the states of Rio Grande do Norte, Paraíba, Pernambuco, Alagoas, Sergipe, and Bahia, particularly in territories and populations with greater social vulnerability. There is a significant proportion of the population living in precarious housing conditions, without access to basic sanitation, with limited access to healthcare [12,13,14,15]. In the other states of the region (Maranhão, Piauí, and Ceará), the transmission of the disease is restricted to small foci [9].

The term/concept of vulnerability, in the sense used in this work, was first proposed in the 1980s, from studies on acquired immunodeficiency syndrome (AIDS), where the evolution of the epidemic was directly associated with the sociodemographic conditions of people infected by HIV. Its adoption enabled the delineation of broader and intersectoral prevention and promotion practices, considering not only the individual dimension, but social and operational aspects of health systems [14]. Similarly, NTDs are intrinsically associated with conditions of poverty and restricted access to diagnosis and treatment. This group of diseases integrates different vulnerability scenarios, where poor living conditions and health inequities represent critical social determination factors for occurrence as public health problems [15,16]. In 2017, the northeast region contained 91.3% of the cases in Brazil [17]. This region was also responsible for 45.7% of hospital admissions and 64.6% of deaths registered in the country [18]. From this perspective, surveillance actions should be strengthened with the integration of data on notifications of cases and schistosomiasis-related hospitalizations.

In schistosomiasis mansoni, as in other NTDs, the costs generated to national health systems are not the only economic impact; there is a dynamic cycle in which the disease acts as both a cause and consequence of poverty, contributing to worsening existing socioeconomic conditions. The potential for schistosomiasis to cause disability and death in affected persons is high, although official records are limited, as is research on the burden of disease in endemic countries. According to the Global Burden of Disease Study 2016, the global burden of schistosomiasis is estimated at 1.9 million disability-adjusted life years (DALYs) [19]. In Brazil, schistosomiasis represents the second leading NTD among those analyzed, after Chagas disease [2,20]. In another study, Nascimento and collaborators, in 2018 and 2019 conducted a study in a northeastern state that showed an average quality of life score of 31.26 QALYs for the population with the chronic digestive forms of schistosomiasis and a total estimated cost per hospitalized case of R$136,087,909.29 (calculation from dollars at the time). It was also estimated at 230,991.75 DALYs, with the majority (219,623—95%) represented by the component of Years of Life with Disability [21].

Considering the strategic nature of this region for the country, the present study aims to analyze the temporal trend and spatial patterns of schistosomiasis morbidity in the northeast region of Brazil in the period 2001–2017.

## 2. Materials and Methods

This is a mixed ecological study, of time series and spatial analyses, based on operational and epidemiological indicators of morbidity for schistosomiasis. The units of analysis were the municipalities and states of the northeast region of Brazil, in the period 2001–2017.

### 2.1. Study Area

The study was performed in the northeast region of Brazil, an endemic area for schistosomiasis, composed of nine states (Maranhão, Piauí, Ceará, Rio Grande do Norte, Paraíba, Pernambuco, Alagoas, Sergipe, and Bahia) (Figure 1). It occupies an area of approximately 1.6 million km^2^ (approximately 18.3% of the national territory), with a population of 53 million (around 28% of the Brazilian population), corresponding to a population density of approximately 33 inhabitants per km^2^ [22]. With 1794 municipalities (almost 32% of the country’s total), it represents one of the most socially vulnerable regions, with a Human Development Index (HDI) of 0.709 and a Social Vulnerability Index (SVI) of 0.306 in 2017 [23].

In Brazil, 83.7% of the inhabitants have access to a treated water supply; however, in the northeast region, this figure is 73.9%. The proportion of households that have access to garbage collection and sewage treatment in Brazil is 54.1%, while, in the northeast, it is 28.6% [24].

### 2.2. Data Sources

In the hospital network, a total of 7063 hospitals were registered in Brazil in 2020, 75.8% (*n* = 5334) being general hospitals, and 28 central public health laboratories (Laboratórios Centrais de Saúde Pública: LACEN). The northeast region ranks second in the number of hospitals (29.8%, *n* = 2094), 74.4% (*n* = 1557) of which are general hospitals, with a total of 9 LACENs (http://tabnet.datasus.gov.br/cgi/deftohtm.exe?cnes/cnv/estabbr.def, accessed on 23 August 2021).

Publicly available secondary data from the Ministry of Health (MoH) Information System of the Schistosomiasis Control Programme (Sistema de Informação do Programa de Controle da Esquistossomose: SISPCE) and the Information System for Notifiable Diseases (Sistema de Informação de Agravos de Notificação: SINAN) were analyzed, both made available by the General Coordination of Surveillance of Zoonoses and Vector-borne Diseases of the Health Surveillance Secretariat of the MoH (CGZV/DEIDT/SVS/MS).

Activities developed in schistosomiasis-endemic areas are registered in the SISPCE, whereas severe cases and possible outbreaks of the disease in SINAN [9]. In areas defined as non-endemic, all schistosomiasis cases should be investigated and notified by the SINAN. The SISPCE was developed and planned to record the program’s control activities in the field, which are developed through parasitological fecal surveys and, therefore, do not record individual data of eventual cases [9]. The technique recommended by the Brazilian Ministry of Health is the quantitative Kato–Katz method for the carrying out of coproscopic surveys for *S. mansoni* eggs. This technique allows the parasite load of the cases to be estimated [9].

We also used data from the Hospital Information System—the Unified Health System (Sistema de Informação Hospitalar—Sistema Único de Saúde: SIH-SUS) through analysis of the Authorization for Hospital Admission (Autorização de Internação Hospitalar: AIH) with a principal or secondary diagnosis of schistosomiasis (B65), according to the International Statistical Classification of Diseases and Health Problems (ICD), in its 10th revision. The hospital admission rate was obtained from (number of admissions due to schistosomiasis divided by the resident population) by 100,000 inhabitants [9].

Population data were obtained from the Brazilian Institute of Geography and Statistics (Instituto Brasileiro de Geografia e Estatística: IBGE), based on the 2000 and 2010 censuses, as well as estimates in intercensal years [22].

### 2.3. Statistical Analyses

The SISPCE and SINAN data were analyzed independently and also integrated, according to the time period and municipalities of residence. The percentage of positivity was calculated from the number of examinations with a positive result for *S. mansoni* in the feces multiplied by 100 and divided by the total number of examinations performed, according to the norms of the MoH and the WHO [9]. This indicator describes how the disease behaves in the territories, in order to classify them according to the epidemiological parameters of the MoH: endemic, focal, vulnerable, or indeterminate. Based on this classification, the strategic control measures are planned and elaborated [9].

The gross schistosomiasis case detection rate (SISPCE and SINAN integration) was calculated by the total number of reported cases of schistosomiasis divided by the resident population of each municipality, multiplied by 100,000.

The municipalities were characterized according to general criteria: 1—residence in the capital (yes or no); 2—municipalities extremely poor (yes or no); 3—municipalities of the semi-arid region (yes or no); 4—Social Vulnerability Index of the Institute for Applied Economic Research (SVI—Instituto de Pesquisas Econômicas Aplicadas (IPEA)): very low (0.000–0.199), low (0.200–0.299), medium (0.300–0.399), high (0.400–0.499), very high (0.500–1.000); 5—Human Development Index (HDI): very low (0.000–0.499), low (0.500–0.599), medium (0.600–0.699), high (0.700–0.799), very high (0.800–1.000); 6—Social Prosperity Index of the IPEA (SPI-IPEA), combination of HDI and SVI; and 7—population size of the municipality: Small Size I (≤20,000 inhabitants), Small Size II (20,001–50,000 inhabitants), Medium Size (50,001–100,000 inhabitants), Large Size (>100,000 inhabitants) [23].

For statistical analysis, the databases were converted and imported into the statistical programme Stata 11.2 (StataCorp. 2009, Release 11, College Station, TX, USA). The epidemiological indicators of morbidity by SINAN and SISPCE and by SIH-SUS were analyzed for the historical series from 2001 to 2017, considering the dimensions of municipalities and state.

Differences in relative frequencies were represented using graphs and tables. Pearson’s chi-square (χ2) test, with calculation of the relative risk (RR) and respective 95% confidence intervals (CI), with a *p*-value greater than 0.05, was used to demonstrate statistical significance.

For time trend analysis of morbidity patterns, Poisson Joinpoint (by inflection points) regression was used. We used the Program Joinpoint Regression version 4.8.0.1 (Joinpoint Regression Program, April 2020; Statistical Methodology and Applications Branch, Surveillance Research Program, National Cancer Institute), with data grouped by states and analyzed according to sociodemographic variables.

The Monte Carlo permutation method was also used to test statistical significance with a view to obtaining the adjustment based on the best line of each segment. Having this definition as principle, the Annual Percentage Variation (APC) and Average Annual Percentage Variation (AAPC) were tested, with their respective 95%CI. The result of inflections of models defined by the program itself, as a criterion for analysis, allows the best representation of the temporal trend. The results obtained show an increase when the APC and AAPC values were positive and statistically significant, a decrease when they were negative and statistically significant, or even no defined trend and no statistical significance throughout the historical series.

For the spatial analyses, distribution maps of cases and hospitalizations for schistosomiasis recorded in the period from 2001 to 2017 were prepared. The time periods defined for analysis were: 2001–2004; 2005–2008; 2009–2012; 2013–2017. The following indicators were calculated: percentage of positivity, crude rate of detection of schistosomiasis cases (SISPCE and SINAN), and rate of hospitalizations for the disease, adjusted for age and sex. The spatial analysis also took into consideration the spatial moving average rate (SMA) and the standardized morbidity ratio (SMR) [25,26]. The SMA was used to identify patterns of concentration of the analyzed rates, considering cases in neighboring municipalities. The SMR sought to identify municipalities with a number of cases above the expected number (excess risk), dividing the number of cases recorded by the expected cases, a technique of non-spatial approach that ignores the effect of spatial autocorrection. The natural breaks method of the Jenks classification algorithm was used to categorize the spatial classes of the crude detection rates, the adjusted hospitalization rates, and the spatial moving average.

In the spatial analysis, the calculation of autocorrelation indicators and the construction of thematic maps were based on the use of the software qGis version 2.18.6 (QGIS Development Team 2017. QGIS Geographic Information System. Open-Source Geospatial Foundation Project. http://qgis.osgeo.org) (accessed on 23 August 2021) and GeoDa version 1.8.16.4.1 (Spatial Analysis Laboratory, University of Illinois 2017, Urbana Champaign, Urbana, IL, USA. http://geodacenter.github.io/ (accessed on 23 August 2021).

### 2.4. Ethical Aspects

As this is a study based on information that is publicly accessible in Brazil, in accordance with Law No. 12,527 of 18 November 2011. As it uses a database whose information is aggregated, without the possibility of identifying any individual data, in accordance with the provisions of the Resolution of the national health council (Conselho Nacional de Saúde: CNS) number 510 of 7 April 2016, a statement was obtained from the Research Ethics Committee of the Federal University of Ceará (Fortaleza, Brazil) exempting the need for evaluation.

## 3. Results

From 2001 to 2017, a total of 15,574,392 parasitological exams were performed (by the standard Kato–Katz method), and 941,961 (6.0%) were positive. Throughout the historical series, positivity rates ranged from 7.2% in 2001 to 3.9% in 2017, a reduction of approximately 46%. In 2001, the states of Pernambuco and Alagoas presented the highest positivity rates, with 25.4% and 16.5%, respectively (Appendix A). The states of Sergipe and Alagoas presented a differentiated pattern, with higher positivity rates (Figure 2A). Paraíba showed a reduction of its positivity from 2015 onwards.

The notification of cases of the disease in SINAN also showed a reduction in the northeast region, particularly from 2008, when the crude detection rate remained without a defined temporal pattern (Figure 2B).

The integrated analysis of SISPCE and SINAN showed a reduction in the crude detection rate in all states, except in the state of Alagoas, which maintained a high rate in 2017 (more than 200 cases per 100,000 inhabitants), as compared to the other states in the region (Figure 2C). Although the highest proportion of cases recorded in SISPCE and SINAN was found in the state of Bahia, with 31.6% (*n* = 328,787) and a crude detection rate of 135.22 cases per 100,000 population (RR = 2.16), the highest crude rate, 492.06 cases per 100,000 inhabitants (RR = 7.84), was found in the state of Alagoas (Table 1).

Living outside the state capital was associated with the highest occurrence of the disease, with 139.34 cases per 100,000 inhabitants (RR = 7.82). Municipalities not considered as extremely poor represented 67.2% (*n* = 699,531) of cases, and, on the other hand, municipalities in the semi-arid region had 21.7% (*n* = 226,394) of cases. Municipalities with a “very high” HDI presented gross rates of 193.34 cases per 100,000 inhabitants (RR = 5.66). The municipalities with “low” HDI were those that presented high rates of detection, 196.78 cases per 100,000 inhabitants (RR = 10.32). The municipalities with “very low” HDI presented the highest detection rates, 197.13 cases per 100,000 inhabitants (RR = 8.91), while the municipalities with “small size I and II” had the highest crude rate of 188.79 (95%CI 186.3 to 191.3) and 180.98 (95%CI 178.7 to 183.3), respectively (Table 1).

Throughout the time series, there was a statistically significant downward trend of the case detection rates in the northeast region, particularly from 2003 to 2017 (APC −13.7 [95%CI −15.5 to −11.9]) and (AAPC −11.5 [95%CI −13.9 to −9.1]). The state of Bahia showed the most evident downward trend from 2001−2017 (APC −19.6 [95%CI −24.6 to −14.3]). There was a tendency for these rates to increase among cases residing in the capital of the states from 2001 to 2005 (APC 30.3 [95%CI 10.6 to 53.5]), but with a tendency for a reduction in the subsequent period, 2005−2017 (APC −17.6 [95%CI −21 to −14.1]). In most municipalities located in the semi-arid region, a downward trend in rates was observed between 2003 and 2017 (APC −12.3 [95%CI −14.2 to −10.3]). Moreover, some coastal cities also showed a decrease in the period 2001–2017 (APC −14.3 [95%CI −17.2 to −11.2]) (Table 2).

Municipalities classified with “low” SVI showed a trend of reduction in rates (APC −17.0 [95%CI −23.1 to −10.4]), as well as those with “very low” HDI (APC −17.4 [95%CI −24.8 to −9.3]), from 2001 to 2017. In municipalities with “low” IPS, there was no significant trend from 2001 to 2003 (APC 22.2 [95%CI −16.7 to 79.1]), but a reduction in the period 2003−2017 (APC −12.8 [95%CI −15.1 to −10.5] and AAPC −10.6 [95%CI −13 to −8.1]). Municipalities of “small size II” showed a greater trend of reduction when compared to the other sizes in 2003−2017 (APC −14.4 [95%CI −16.1 to −12.7] and AAPC −12.2 [95%CI −14.5 to −9.7]) (Table 2).

Spatial analysis demonstrated that gross detection rates, SMA, and SMR showed a focal pattern on the coast and in the Zona da Mata of Pernambuco, Alagoas, and Sergipe, while, in areas of Central, Eastern, and Western Bahia, this pattern was reduced over the historical series, leaving some limited areas associated with a higher risk for the disease in the state (Figure 3).

A total of 5879 schistosomiasis-related hospitalizations were recorded from 2001 to 2017 in the northeast region, with 2378 (40.4%) in the state of Pernambuco alone. For the region, the crude rate was 0.64 hospitalizations related to schistosomiasis per 100,000 inhabitants (Table 1), ranging from 0.82 (2001) to 0.09 (2017) hospitalizations per 100,000 inhabitants, a reduction of 89%. The state of Alagoas showed the highest crude rate of 1.72 (95%CI 1.27 to 2.18) and RR (16.50, 95%CI 7.50 to 36.26), with a statistically significant reduction trend from 2001 to 2017 (AAPC −21.8 [95%CI −24.7 to −18.8]) (Table 2). The analysis of the SIH-SUS records also indicates a reduction in the rate of hospitalization for schistosomiasis adjusted by age and sex, also highlighting the state of Alagoas, which varied from 3.04 hospitalized cases per 100,000 hospitalizations in 2001 to 0.13 in 2017, a reduction of approximately 96% (Figure 2D).

A reduction in the schistosomiasis hospitalization rates was also observed in the spatial analysis. The adjusted hospitalization rate showed greater concentration on the coast and in the savannah areas of the states of Paraíba, Pernambuco, Alagoas, and Southern Sergipe, with reductions over the periods considered. The same was observed in the distribution maps of the SMA and SMR, with a reversal of the initial patterns of spatial concentration over time, maintaining it mainly in focal areas of the Pernambuco coast and in some focal areas of the state of Bahia, with no more expression in Piauí (Figure 4).

## 4. Discussion

This integrated assessment of schistosomiasis-related morbidity in the northeast region of Brazil during a 17-year period of analysis revealed the persistence of high incidence of the disease, despite the reduction of indicators related to case notification and hospitalization observed in the official health information systems. This period is characterized by a systematic decrease in technical surveillance and control actions, which probably compromised state programs operationally and may have contributed to the reduced figures in these epidemiological analyses. In this context, the maintenance of focal areas is worrisome and shows the intensity of the endemic disease, particularly in the last five years, requiring an intensification of surveillance, control, and healthcare actions [7,8]. The severe political–institutional and social crisis that the country is currently going through, particularly since 2016, may have contributed to the intensification of schistosomiasis morbidity and mortality in more recent years [18,19,27,28].

The observed reduction in positivity rates corroborates the findings of other studies and may possibly be explained by irregularities in the implementation of the surveillance program, favored by the reduced operational structure in the states and municipalities, and also the occurrence of public health emergencies, diverting attention from the disease [27,28]. A study conducted in the state of Alagoas evidenced that, in 2017, there was a higher proportion of cases of the disease when compared to the total recorded in Brazil (7023 and 21,962, respectively), indicating the maintenance of the severity of the endemic in the state [27,29,30]. From another perspective, the results of the National Survey of Prevalence of Schistosomiasis and Geo-helminthiasis (INPEG) indicated that the state of Sergipe showed considerable positivity rates (8.18%), demonstrating the need for political prioritization [28,31].

Historically, the state of Pernambuco has remained with high endemicity for schistosomiasis, highlighting the implementation in 2011 of the Program for Confronting Neglected Diseases (SANAR) [32,33]. This program prioritized the practice of integrated approaches of Health Surveillance with Primary Health Care (PHC) allied to mass treatment in areas hyperendemic for schistosomiasis, in accordance with WHO recommendations [33]. These strategies resulted in increased early detection of cases, as well as treatment and follow-up of cases aiming at reducing morbidity and mortality. Our study also showed that the strategy of mass treatment in hyperendemic areas promoted an effective reduction of the occurrence of the disease in Pernambuco, but the other hyperendemic states have not yet been able to advance on this WHO recommendation [32].

The registration of cases in the SISPCE and SINAN, in the municipalities of the northeast region, remains with limitations because some municipalities have not systematically performed the recommended stool investigations in areas considered to be at risk. This fact is due, in part, to the results of the inadequate decentralization of endemic disease control actions (1999–2000), in the context of the municipalization of the stage of performing stool exams and subsequent computerization of the results [28,34].

SISPCE reflects how the PCE was initially structured in the municipalities, having its origin based on a centralized and verticalized structure, which has always guided the way of acting in past decades, providing only operational data and adding little information for the production of epidemiological indicators [28]. In this sense, there is a need for its restructuring to adapt to new methods and techniques of action with the PHC, as well as recognition of these areas from the registration of cases and the treatment undertaken, integrating the attention to health surveillance [28,34]. The strategy of this study to integrate the databases of these information systems aimed to increase the sensitivity to capture cases of schistosomiasis in different scenarios of endemicity in the northeast region.

The reduction in the number of exams performed was notorious in the period analyzed, which may have contributed to underreporting, influencing the epidemiological interpretations by biasing the real temporal trends of the disease [7,35]. We believe that outbreaks of dengue or other arboviruses, and, more recently, the pandemic caused by SARS-CoV-2, overburden health services and compromise the development of schistosomiasis surveillance and control activities [7,35,36,37]. It was observed, in general, relative similarity between SISPCE and SINAN in terms of the trend of reduction in schistosomiasis cases in the historical series, although the state of Alagoas remained in evidence, with an expressive rate of detection of the disease among the states.

Living outside the state capital was associated with higher occurrence of the disease, reinforcing the typical focal and rural nature of schistosomiasis. However, other studies have indicated that the urbanization process has become an increasingly important determining factor for the occurrence of different parasitic diseases, due to the expansion of poverty in the suburban areas, inordinate occupation of territories, population densification, precarious constructions, and inadequate sanitation conditions, facts that together contribute to the maintenance of transmission in focal areas and its expansion to other areas [38].

The municipalities of the semi-arid region presented a lower risk for the occurrence of the disease in this study, although the MoH considers these areas to be of higher potential risk for transmission due to the presence of planorbids as intermediate hosts, mollusks of the genus *Biomphalaria* [13,39]. In these areas, favorable conditions for the establishment of snail populations from other endemic regions are recorded. A more recent example refers to areas of influence of the project to transpose the waters of the São Francisco River, an intervention that may favor the translocation of risks and the occurrence of transmission, exposing local communities to the disease [13,39].

The higher occurrence of schistosomiasis in small, extremely poor municipalities, with very high SVI, low HDI, and very low SPI, reflects the strong association between the occurrence of schistosomiasis in endemic areas and the critical social determinants of poverty and extreme poverty, missing basic sanitation, limited access to health services, low education, and low quality of life [40,41]. These and other unfavorable aspects for neglected populations in these territories also favor the permanence of other NTDs that can overlap, particularly those whose transmission is related to limited access to quality water, coupled with a lack of health education and hygiene actions [15,41]. Given this context, a strong association has been systematically recorded between morbidity and mortality indicators of the disease and socio-demographic, cultural, and economic risk factors, making the population more vulnerable to infection by *S. mansoni*, clinical progression, or evolution to death [41].

There was a statistically significant reduction in the temporal trend of case detection, particularly in the state of Bahia, corroborating the findings of the study by Silva et al. (2019) [8]. On the other hand, there was an increase in the trend of occurrence of cases in the state capital (2001–2005), which may be associated with the migratory movement of cases from rural to urban areas [36]. The forms with more severe clinical syndrome are still conditioned to repetitive and intense contact with sources containing *S. mansoni*, a condition quite common in inland areas of the northeast region, where there is the presence of contaminated water bodies, a fact that also includes the possibility of the occurrence of infections through rural tourism [42,43,44].

The analysis of spatial distribution patterns indicated a higher concentration of cases and hospital admissions for schistosomiasis in areas along the coast of the states of Pernambuco, Alagoas, and Sergipe, corroborating other studies that revealed high morbidity and mortality rates due to the disease and the formation of clusters in these areas [30,45,46,47,48]. Hospital admissions due to schistosomiasis result from the evolution of the disease, when there were failures in the identification of cases, timely treatment, and healthcare of the affected person [8]. The higher concentration of hospital admissions of people from coastal areas may be related to the increased migration of people from rural areas (endemic) to urban areas, which, although there are more opportunities for work and employment, also suffer from serious problems of urban infrastructure and a lack of basic sanitation, which favors the installation of precarious housing without adequate sanitary infrastructure, and the formation of pockets of poverty, increasing the expansion of the disease [40]. Analyses performed with different tools (positivity rate, gross detection rate, spatial moving average rate “SMA”, and standardized morbidity ratio “SMR”) show that there is homogeneity in the spatial distribution of schistosomiasis, remaining in the same geographical spaces over time, although it presents with a focal characteristic [8,28,40,49].

The trends verified in the time series analysis indicate a reduction in the hospitalization rate in the northeast region, which corroborates the studies of Resendes (2005) and Silva (2019) [8,48]. The state of Pernambuco showed the highest proportion of hospitalizations in the historical series, a fact also verified in the study by Barbosa (2016), when the state accounted for 25.2% of the hospitalizations recorded in all of Brazil in the period 2008–2014 [50].

During the period of this study, the Brazilian MoH recommended the performance of control actions with positivity rates above 50%. Despite the current recommendations, the states of the northeast region had technical–operational limitations for development, except in the state of Pernambuco, which established, in 2011, a state policy to prioritize the control of NTDs, called the SANAR Program (Integrated Plan of Actions for Confronting Neglected Diseases), in which schistosomiasis, among eight other NTDs, was defined as a priority. A study conducted by Facchini and collaborators in 2018 found that the political decision made by the state of Pernambuco to implement the SANAR Program in 2011 impacted the reduction of the schistosomiasis burden. This program was effective in reducing the occurrence of the disease in hyperendemic areas in this state, with a more consistent operational response in areas with two cycles of collective treatment [32].

It is important to highlight that schistosomiasis is a disease that can also generate stigma, besides causing physical disability, reduced quality of life, and evolution to death [8]. Thus, its early diagnosis is strategic, together with comprehensive care to the affected person in order to prevent the progression of the disease to more complex clinical forms [8,32,51].

Globally, approximately 250 million people develop schistosomiasis, resulting in 1430,000 DALYs (disability-adjusted life years) per year. Despite advances, there are critical limitations in the science resulting in a restriction of candidate schistosomiasis vaccines (Sm28GST/Sh28GST, Sm-p80, Sm14, and Sm-TSP-1/SmTSP-2) reaching the different phases of clinical trials. Thus, increasing efforts are needed to achieve the WHO targets set for NTD control [52]. 

The present study presents limitations inherent to the databases used for the analysis. The high number of cases with incomplete or poor-quality records in some fields may have compromised the analyses. In addition, operational issues related to the differential performance of the municipalities in the surveillance and control of the disease may have influenced the trends observed. The approach to hospital admissions in an integrated manner with traditional epidemiological indicators points in the direction of “control”.

However, it is known that hospitalization is generally associated with events associated with the evolution of the disease and, therefore, schistosomiasis is not recorded as the initial cause of hospitalization. Despite these limitations, the integration strategies used, the territorial reach of a priority region for control in Brazil, together with the study of a 17-year historical series, bring into perspective the relevance of the study [53].

## 5. Conclusions

The northeast region of Brazil remains a priority endemic area for schistosomiasis control throughout the 17 years analyzed. Despite the reduction in case detection and hospitalizations, the persistence of focal areas of the disease in coastal areas is recognized. This reduction may signal a possible positive impact of control on epidemiological patterns, but still below the pre-established objectives. Reduced surveillance and control actions in the period evidence the need for strengthening and qualification in these focal areas identified in the states of Pernambuco, Alagoas, and Sergipe.

As a disease strongly associated with conditions of social vulnerability, the return of the growth of poverty and extreme poverty, reflected by increasing social inequalities in this region of the country since 2016, is of great concern. This fact reinforces the need to monitor the trends verified by studies developed in the region to confirm temporal trends and spatial patterns. The considerable reduction in public investments in the country can make the process of the surveillance and control of NTDs even more complex and open space for the expansion of transmission in the areas identified by this study. Therefore, it is important to emphasize and strengthen the Unified Health System (SUS) for the sustainability of actions to control Schistosomiasis and other NTDs in Brazil.

## Figures and Tables

**Figure 1 tropicalmed-06-00193-f001:**
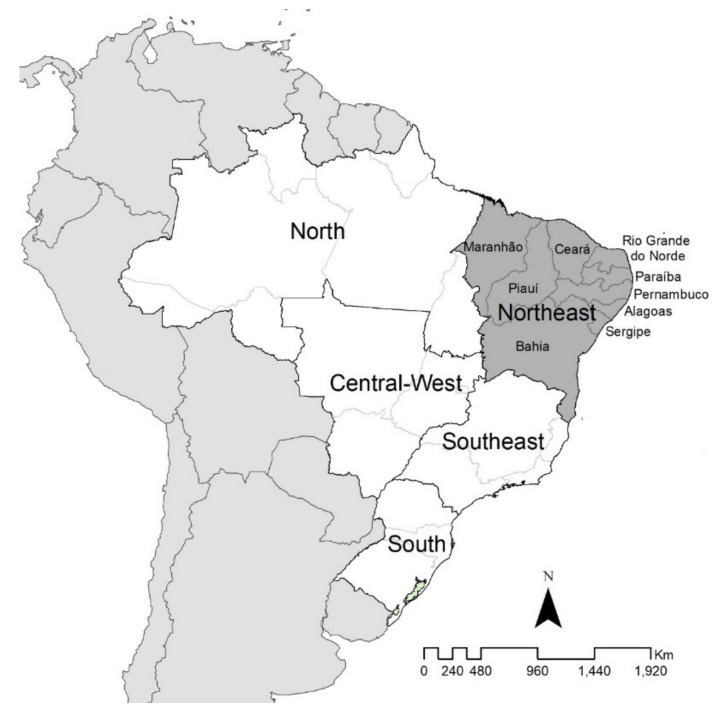
Areas of study: states and northeast regions of Brazil.

**Figure 2 tropicalmed-06-00193-f002:**
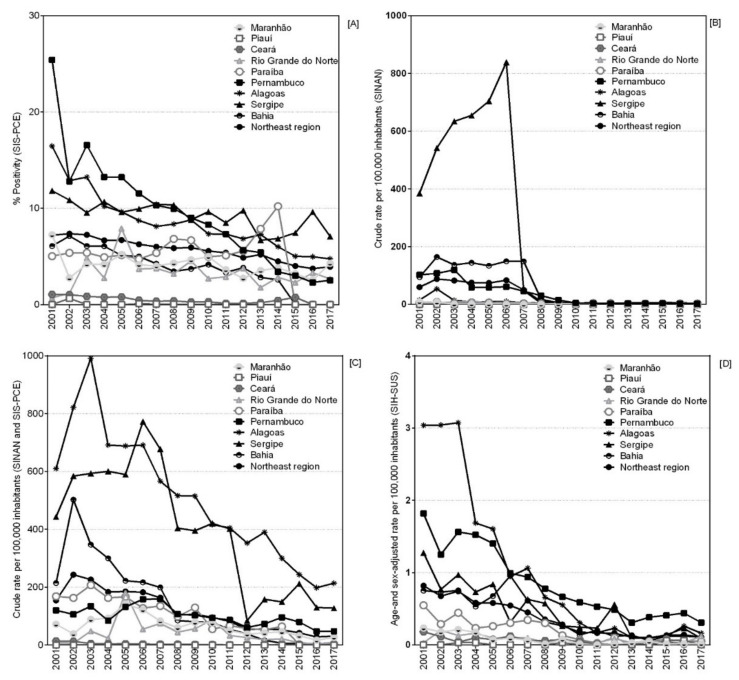
Percentage of positivity (**A**), crude case detection rate (SINAN) (**B**), crude case detection rate (SINAN and SISPCE) (**C**) and rate of hospital admissions (**D**) per 100,000 Inhabitants—according to year and state of the Northeast region. Brazil, 2001–2017.

**Figure 3 tropicalmed-06-00193-f003:**
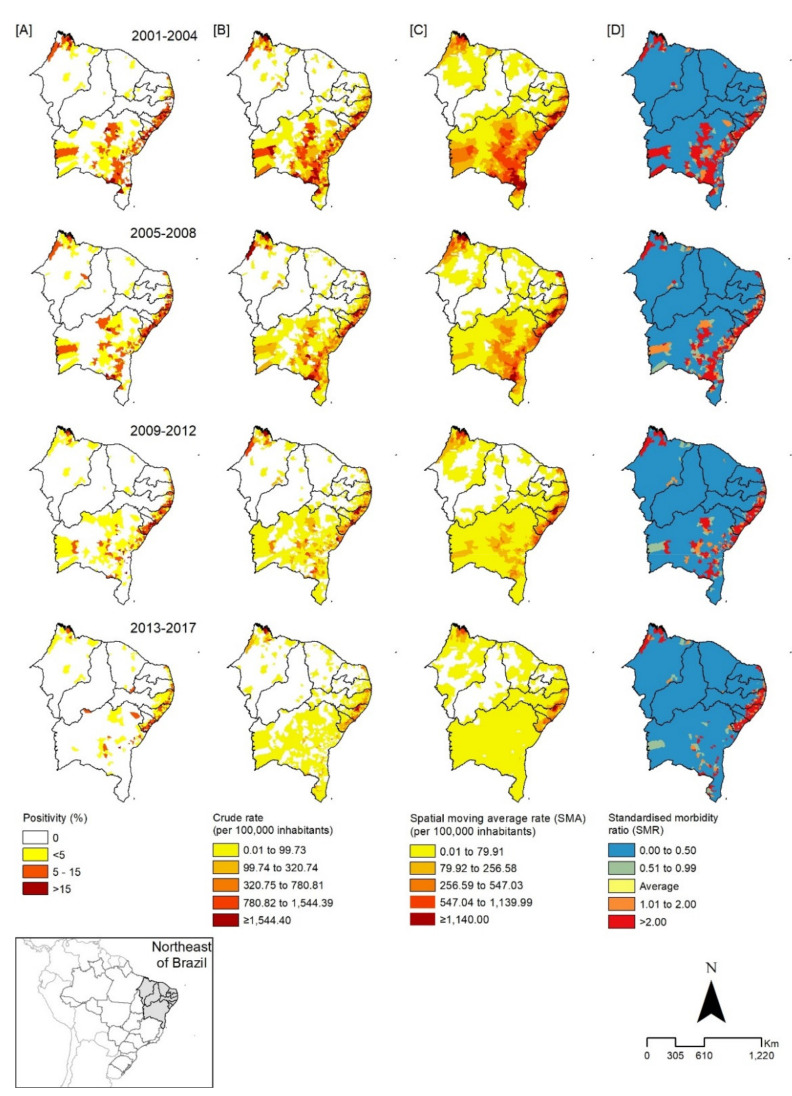
Spatial distribution of schistosomiasis cases according to percentage of positivity (**A**), crude detection rate (**B**), spatial moving average rate (SMA) of detection (**C**), and standardized morbidity ratio (SMR) of detection (**D**), in municipalities of the states of the northeast Region, divided by trienniums, Brazil, 2001–2017.

**Figure 4 tropicalmed-06-00193-f004:**
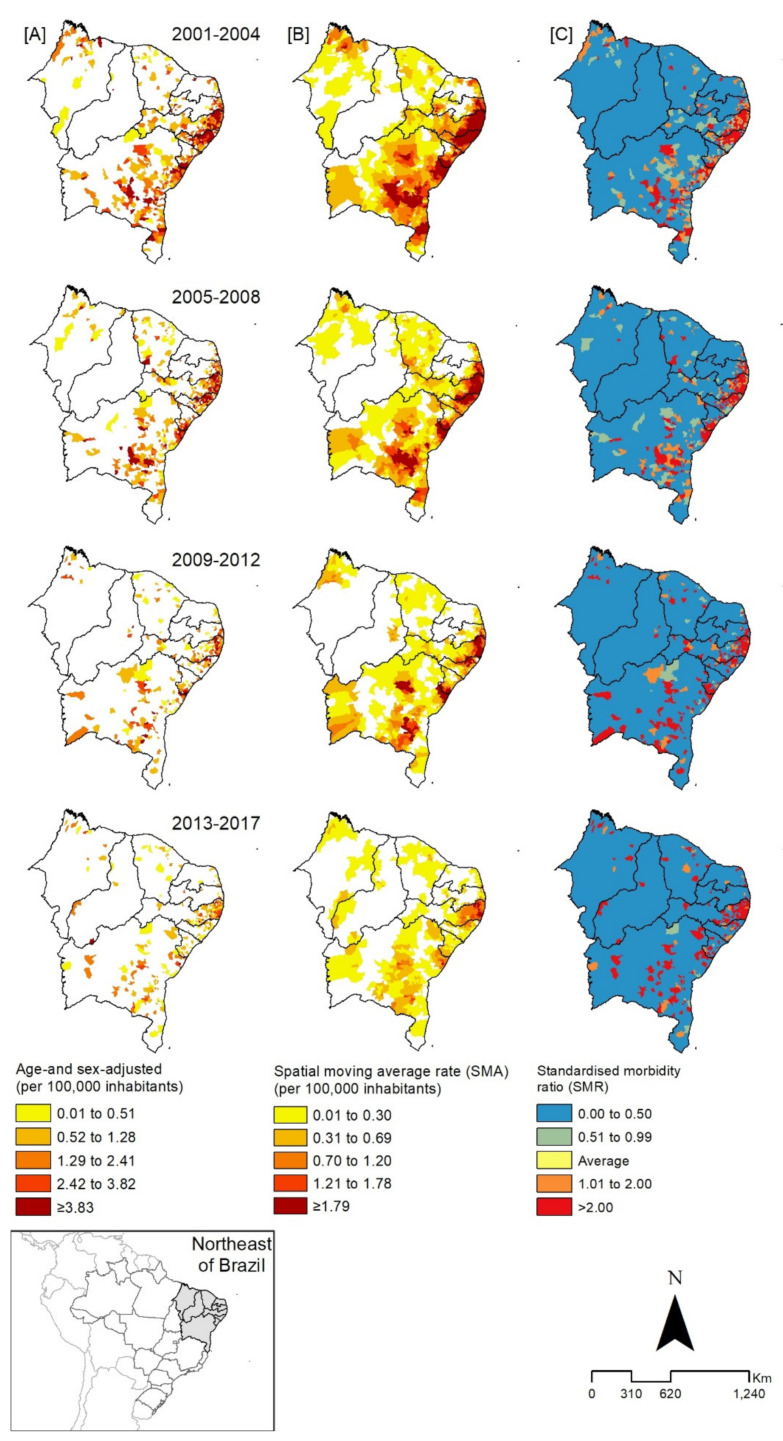
Spatial distribution of hospitalization for schistosomiasis according to analyses by adjusted hospitalization rate (**A**), spatial moving average rate (SMA) (**B**), and standardized morbidity ratio (SMR) (**C**), in municipalities of the states of the northeast region, according to triennials, Brazil, 2001–2017.

**Table 1 tropicalmed-06-00193-t001:** Crude rate case detection (SINAN and SISPCE) and crude rate hospital admissions for schistosomiasis (per 100,000 inhabitants) according to sociodemographic variables in the northeast region, Brazil, 2001–2017.

Variable	SISPCE + SINAN				SIH			
	*n* (%)	Crude Rate (95%CI)	RR (95%CI)	*p*-Value	*n* (%)	Crude Rate (95%CI)	RR (95%CI)	*p*-Value
Total	1,040,983 (100.0)	112.95 (112.10–113.80)			6030 (100.0)	0.65 (0.59–0.72)	-	
State of residence								
Maranhão	71,275 (6.8)	62.48 (60.60–64.37)	1.00		117 (1.9)	0.10 (0.03–0.18)	1.00	
Piauí	35 (0.0)	0.06 (0.00–0.15)	0.00 (0.00–0.00)	<0.0001	14 (0.2)	0.03 (0.00–0.09)	0.30 (0.04–2.44)	0.2605
Ceará	4,680 (0.4)	3.19 (2.80–3.57)	0.05 (0.00–0.06)	<0.0001	170 (2.8)	0.12 (0.04–0.19)	1.11 (0.42–2.92)	0.8299
Rio Grande do Norte	22,154 (2.1)	40.22 (38.00–42.40)	0.64 (0.60–0.69)	<0.0001	67 (1.1)	0.12 (0.00–0.24)	1.18 (0.35–4.04)	0.7880
Paraíba	61,902 (5.9)	94.80 (91.70–97.88)	1.52 (1.50–1.59)	<0.0001	302 (5.0)	0.47 (0.25–0.69)	4.49 (1.88–10.76)	0.0007
Pernambuco	148,575 (14.3)	97.10 (95.10–99.14)	1.55 (1.50–1.61)	<0.0001	2508 (41.6)	1.64 (1.38–1.91)	15.76 (7.39–33.64)	<0.0001
Alagoas	267,328 (25.7)	492.06 (484.40–499.70)	7.84 (7.60–8.11)	<0.0001	937 (15.5)	1.72 (1.27–2.18)	16.5 (7.51–36.23)	<0.0001
Sergipe	136,247 (13.1)	380.17 (371.90–388.50)	6.07 (5.80–6.30)	<0.0001	282 (4.7)	0.81 (0.42–1.19)	7.73 (3.21–18.64)	<0.0001
Bahia	328,787 (31.6)	135.22 (133.30–137.10)	2.16 (2.10–2.24)	<0.0001	1633 (27.1)	0.67 (0.54–0.81)	6.43 (2.99–13.86)	<0.0001
Residence in the capital								
No	1,005,388 (96.6)	139.34 (138.20–140.50)	7.82 (7.50–8.17)	<0.0001	5135 (85.2)	0.71 (0.63–0.79)	1.58 (1.18–2.12)	0.0021
Yes	35,595 (3.4)	17.79 (17.00–18.55)	1.00		895 (14.8)	0.45 (0.33–0.57)	1.00	
Municipality extremely poor								
No	699,531 (67.2)	108.49 (107.40–109.50)	1.00		4184 (69.4)	0.65 (0.57–0.73)	0.97 (0.77–1.21)	0.7858
Yes	341,452 (32.8)	123.31 (121.60–125.00)	1.14 (1.10–1.16)	<0.0001	1846 (30.6)	0.67 (0.54–0.79)	1.00	
Municipality of the semi-arid region								
No	814,589 (78.3)	147.84 (146.50–149.20)	1.00		4683 (77.7)	0.85 (0.75–0.95)	2.34 (1.82–3.01)	<0.0001
Yes	226,394 (21.7)	61.08 (60.00–62.11)	0.41 (0.40–0.42)	<0.0001	1347 (22.3)	0.36 (0.28–0.44)	1.00	
SVI								
Very low	4 (0.0)	-			1 (0.0)	-	-	-
Low	30,247 (2.9)	34.10 (32.50–35.69)	1.00		155 (2.6)	0.17 (0.06–0.29)	1.00	
Medium	159,300 (15.3)	45.84 (44.90–46.77)	1.34 (1.30–1.41)	<0.0001	1819 (30.2)	0.52 (0.42–0.62)	3.03 (1.54–5.99)	0.0014
High	500,586 (48.1)	164.69 (162.80–166.60)	4.82 (4.60–5.06)	<0.0001	2873 (47.6)	0.95 (0.80–1.09)	5.48 (2.80–10.71)	<0.0001
Very high	350,846 (33.7)	193.34 (190.70–196.00)	5.66 (5.40–5.94)	<0.0001	1182 (19.6)	0.66 (0.50–0.81)	3.80 (1.90–7.61)	0.0002
HDI								
Very low	1,522 (0.1)	48.97 (38.90–59.09)	2.57 (2.10–3.17)	<0.0001	6 (0.1)	-	-	-
Low	553,516 (53.2)	196.78 (194.60–198.90)	10.32 (9.90–10.71)	<0.0001	2245 (37.2)	0.80 (0.66–0.93)	1.73 (1.30–2.30)	0.0001
Medium	433,275 (41.6)	120.16 (118.70–121.60)	6.31 (6.10–6.55)	<0.0001	2501 (41.5)	0.69 (0.58–0.81)	1.50 (1.14–1.99)	0.004
High	52,670 (5.1)	19.04 (18.40–19.71)	1.00		1278 (21.2)	0.46 (0.36–0.57)	1.00	
SPI								
Very low	541,034 (52.0)	197.13 (195.00–199.30)	8.91 (8.60–9.29)	<0.0001	2216 (36.7)	0.81 (0.67–0.94)	4.24 (2.16–8.33)	<0.0001
Low	317,647 (30.5)	154.90 (152.70–157.10)	7.01 (6.70–7.31)	<0.0001	1696 (28.1)	0.83 (0.67–0.99)	4.36 (2.21–8.63)	<0.0001
Medium	123,410 (11.9)	71.28 (69.60–72.92)	3.23 (3.10–3.38)	<0.0001	1010 (16.7)	0.58 (0.43–0.73)	3.05 (1.51–6.15)	0.0018
High	41,618 (4.0)	22.08 (21.20–22.95)	1.00		960 (15.9)	0.51 (0.37–0.64)	2.66 (1.31–5.37)	0.0065
Very high	17,274 (1.7)	21.45 (20.10–22.77)	0.97 (0.90–1.05)	0.4407	148 (2.5)	0.19 (0.07–0.31)	1.00	
Size of municipality								
Small Size I	366,294 (35.2)	188.79 (186.30–191.30)	1.62 (1.60–1.66)	<0.0001	944 (15.7)	0.49 (0.36–0.62)	1.00	
Small Size II	397,032 (38.1)	180.98 (178.70–183.30)	1.55 (1.50–1.59)	<0.0001	1997 (33.1)	0.91 (0.74–1.07)	1.85 (1.34–2.54)	0.0002
Medium Size	155,869 (15.0)	116.45 (114.10–118.80)	1.00		1024 (17.0)	0.76 (0.57–0.95)	1.55 (1.08–2.24)	0.0178
Large Size	121,788 (11.7)	32.53 (31.80–33.28)	0.28 (0.30–0.29)	<0.0001	2065 (34.2)	0.55 (0.45–0.65)	1.12 (0.82–1.54)	0.4840

-, not calculated; N, number; %, percentage; 95%CI: 95% Confidence Interval. Size of municipality: Small Size I: ≤20,000 inhabitants, Small Size II: 20,001–50,000 inhabitants, Medium Size: 50,001–100,000 inhabitants, Large Size: >100,001 inhabitants. SISPCE: Schistosomiasis Control Program Information System (Sistema de Informação do Programa de Controle da Esquistossomose). SINAN: Information System of Notifiable Diseases (Sistema de Informação de Agravos de Notificação). SIH-SUS: Hospital Information System—Unified Health System (Sistema de Informação Hospitalar—Sistema Único de Saúde). SVI: Social Vulnerability Index. HDI: Human Development Index. SPI: Social Prosperity Index.

**Table 2 tropicalmed-06-00193-t002:** Annual Percentage Change (APC) and Average Annual Percentage Change (AAPC) of schistosomiasis case detection and hospitalization for the disease according to sociodemographic variables in the northeast region, Brazil, 2001–2017.

Variable	SISPCE + SINAN			SIH		
	Period	APC (95%CI)	AAPC (95%CI)	Period	APC (95%CI)	AAPC (95%CI)
Total	2001–2003	19.8 (−10.9 to 61.1)	−11.5 * (−13.9 to −9.1)	2001–2014	−14.2 * (−16.8 to −11.6)	−13.2 * (−15.0 to −11.3)
	2003–2017	−13.7 * (−15.5 to −11.9)		2014–2017	0.8 (−35.2 to 56.9)	
State of residence						
Maranhão	2001–2005	22.5 (0 to 50.2)	−7.6 * (−11.8 to -3.1)	2001–2017	−17.5 * (−20.9 to −14.0)	−17.5 * (−20.9 to −14.0)
	2005–2017	−13.1 * (−16.9 to −9)				
Piauí	2001–2017	−3.1 (−12.1 to 6.8)	−3.1 (−12.1 to 6.8)	2001–2017	8.8 (−0.7 to 19.2)	8.8 (−0.7 to 19.2)
Ceará	2001–2017	−19.4 * (−22.8 to −15.8)	−19.4 * (−22.8 to −15.8)	2001–2017	−9.8 * (−13.2 to −6.1)	−9.8 * (−13.2 to −6.1)
Rio Grande do Norte	2001–2005	110.5 (−13.5 to 412.6)	−12.0 * (−21.9 to −0.8)	2001–2017	−7.8 * (−12.9 to −2.4)	−7.8 * (−12.9 to −2.4)
	2005–2017	−20.1 * (−28.8 to −10.3)				
Paraíba	2001–2014	−9.5 * (−12.7 to −6.3)	−10.4 * (−13.9 to −6.7)	2001–2017	−13.8 * (−18.0 to −9.5)	−13.8 * (−18.0 to −9.5)
	2014–2017	−70.6 (−96.1 to 118.8)				
Pernambuco	2001–2017	−4.8 * (−7.5 to −2.1)	−4.8 * (−7.5 to −2.1)	2001–2017	−9.8 * (−11.6 to −8.0)	−9.8 * (−11.6 to −8.0)
Alagoas	2001–2003	18.5 (−5.4 to 48.4)	−8.2 * (−9.9 to −6.4)	2001–2017	−21.4 * (−24.2 to −18.4)	−21.4 * (−24.2 to −18.4)
	2003–2017	−9.8 * (−11 to −8.6)				
Sergipe	2001–2006	8.1 (−6.1 to 24.5)	−9.0 * (-12.8 to −4.9)	2001–2017	−12.5 * (−15.7 to −9.2)	−12.5 * (−15.7 to −9.2)
	2006–2017	−15.5 * (−20.2 to −10.5)				
Bahia	2001–2017	−19.6 * (−24.6 to −14.3)	−19.6 * (−24.6 to −14.3)	2001–2006	−2.8 (−15.0 to 11.1)	−14.2 * (−18.3 to −9.9)
				2006–2017	−20.6 * (−26.4 to −14.4)	
Residence in the capital						
No	2001–2003	18.7 (−11.1 to 58.5)	−11.6 * (−13.9 to −9.2)	2001–2005	−6.7 (−19.0 to 7.4)	−12.6 * (−14.6 to −10.4)
	2003–2017	−13.7 * (−15.5 to −11.9)		2005–2017	−14.4 * (−18.0 to −10.7)	
Yes	2001–2005	30.3 * (10.6 to 53.5)	−9.9 * (−15.2 to −4.2)	2001–2017	−16.8 * (−18.4 to −15.1)	−16.8 * (−18.4 to −15.1)
	2005–2017	−17.6 * (−21 to −14.1)				
Municipality extremely poor						
No	2001–2003	21.3 (−10.5 to 64.4)	−10.2 * (−12.4 to −8)	2001–2017	−12.6 * (−14.1 to −11.1)	−12.6 * (−14.1 to −11.1)
	2003–2017	−12.3* (−14.1 to −10.5)				
Yes	2001–2017	−14.3 * (−17.2 to −11.2)	−14.3 * (−17.2 to −11.2)	2001–2006	−2.2 (−14.7 to 12.2)	−14.1 * (−18.3 to −9.7)
				2006–2017	−20.7 * (−26.6 to −14.2)	
Municipality of the semi-arid region						
No	2001–2003	27.3 (−10.3 to 80.6)	−10.0 * (−12.4 to −7.5)	2001–2005	−7.6 (−19.8 to 6.5)	−13.8 * (−16.0 to −11.6)
	2003–2017	−12.3 * (−14.2 to −10.3)		2005–2017	−15.9 * (−19.7 to −12.0)	
Yes	2001–2017	−18.1 * (−21.1 to −14.9)	−18.1 * (−21.1 to −14.9)	2001–2014	−13.2 * (−15.1 to −11.1)	−11.1 * (−13.0 to −9.3)
				2014–2017	12.8 (−17.9 to 55.0)	
SVI						
Very low	2001–2017	−10.8 * (−16.8 to −4.3)	−10.8 * (−16.8 to −4.3)	2001–2017	−2.6 * (−4.9 to −0.1)	−2.6 * (−4.9 to −0.1)
Low	2001–2017	−17.0 * (−23.1 to −10.4)	−17.0 * (−23.1 to −10.4)	2001–2017	−11.2 * (−14.3 to −8.0)	−11.2 * (−14.3 to −8.0)
Medium	2001–2005	0.4 (−15.5 to 19.3)	−13.0 * (−16.1 to −9.8)	2001–2011	−17.0 * (−19.3 to −14.7)	−14.8 * (−16.3 to −13.2)
	2005–2017	−17.4 * (−21.6 to −12.9)		2011–2017	−6.3 (−15.9 to 4.4)	
High	2001–2003	19.6 (−15.6 to 69.3)	−11.5 * (−13.9 to −9)	2001–2005	−1.2 (−16.9 to 17.4)	−11.7 * (−14.5 to −8.8)
	2003–2017	−13.7 * (−15.9 to −11.5)		2005–2017	−14.8 * (−18.8 to −10.5)	
Very high	2001–2003	23.3 (−10.1 to 69.1)	−10.1 * (−12.3 to −7.8)	2001–2017	−14.2 * (−16.6 to −11.7)	−14.2 * (−16.6 to −11.7)
	2003–2017	−12.2 * (−13.9 to −10.4)				
HDI						
Very low	2001–2017	−17.4 * (−24.8 to −9.3)	−17.4 * (−24.8 to −9.3)	2001–2017	−3.4 (−6.8 to 0.1)	−3.4 (−6.8 to 0.1)
Low	2001–2003	22 (−8.7 to 63.1)	−11.2 * (−13.6 to −8.7)	2001–2006	−4.5 (−17.9 to 11.1)	−14.4 * (−18.3 to −10.2)
	2003–2017	−13.4 * (−15.1 to −11.7)		2006–2017	−19.9 * (−26.4 to −12.9)	
Medium	2001–2006	−2.7 (−10.7 to 5.9)	−11.7 * (−14.1 to −9.3)	2001–2017	−11.5 * (−13.0 to −9.9)	−11.5 * (−13.0 to −9.9)
	2006–2017	−16.1 * (−19.6 to −12.6)				
High	2001–2006	7.8 (−4.1 to 21.1)	−10.6 * (−14.7 to −6.4)	2001–2010	−17.8 * (−21.6 to −13.9)	−13.6 * (−15.9 to −11.3)
	2006–2017	−17.9 * (−22.2 to −13.4)		2010–2017	−3.4 (−13.9 to 8.3)	
SPI						
Very low	2001–2003	20.6 (−9.4 to 60.7)	−11.2 * (−13.5 to −8.8)	2001–2006	−4.5 (−18.1 to 11.4)	−14.4 * (−18.3 to −10.2)
	2003–2017	−13.3 * (−15 to −11.6)		2006–2017	−19.9 * (−26.4 to −12.8)	
Low	2001–2003	22.2 (−16.7 to 79.1)	−10.6 * (−13 to −8.1)	2001–2017	−10.5 * (−12.4 to −8.5)	−10.5 * (−12.4 to −8.5)
	2003–2017	−12.8 * (−15.1 to −10.5)				
Medium	2001–2017	−13.3 * (−16.1 to −10.5)	−13.3 * (−16.1 to −10.5)	2001–2017	−11.7 * (−13.6 to −9.7)	−11.7 * (−13.6 to −9.7)
High	2001–2017	−14.9 * (−20.4 to −9.1)	−14.9 * (−20.4 to −9.1)	2001–2009	−20.0 * (−23.9 to −15.8)	−15.8 * (−18.0 to −13.6)
				2009–2017	−8.1 (−16.1 to 0.5)	
Very high	2001–2017	−12.5 * (−17 to −7.7)	−12.5 * (−17 to −7.7)	2001–2017	−11.1 * (−14.5 to −7.5)	−11.1 * (−14.5 to −7.5)
Size of municipality						
Small Size I	2001–2006	−0.1 (−8.5 to 9.1)	−10.6 * (−13.2 to −8)	2001–2014	−16.3 * (−19.1 to −13.4)	−13.6 * (−16.2 to −11.0)
	2006–2017	−15.5 * (−18.9 to −11.9)		2014–2017	19.2 (−31.7 to 108.0)	
Small Size II	2001–2003	18.6 (−9 to 54.6)	−12.2 * (−14.5 to −9.7)	2001–2006	−1.7 (−14.9 to 13.7)	−13.9 * (−18.1 to −9.5)
	2003–2017	−14.4 * (−16.1 to −12.7)		2006–2017	−20.3 * (−26.4 to −13.8)	
Medium Size	2001–2017	−11.3 * (−14.1 to −8.4)	−11.3 * (−14.1 to −8.4)	2001–2017	−11.6 * (−13.6 to −9.7)	−11.6 * (−13.6 to −9.7)
Large Size	2001–2007	−2.8 (−8.4 to 3.1)	−11.5 * (−14.4 to −8.5)	2001–2010	−15.7 * (−18.7 to −12.7)	−12.6 * (−14.3 to −10.8)
	2007–2017	−18.2 * (−22.2 to −14)		2010–2017	−5.3 (−12.9 to 2.9)	

* Significantly different from 0 (*p* < 0.05). Monte Carlo permutation method; 95%CI: 95% Confidence Interval. Size of municipality: Small Size I: ≤20,000 inhabitants, Small Size II: 20,001–50,000 inhabitants, Medium Size: 50,001–100,000 inhabants, Large Size: >100,001 inhabitants. APC: Annual Percentage Change; AAPC: Average Annual Percentage Change; SISPCE: Schistosomiasis Control Program Information System (Sistema de Informação do Programa de Controle da Esquistossomose). SINAN: Information System of Notifiable Diseases (Sistema de Informação de Agravos de Notificação). SIH-SUS: Hospital Information System—Unified Health System (Sistema de Informação Hospitalar—Sistema Único de Saúde); SVI: Social Vulnerability Index; HDI: Human Development Index; SPI: Social Prosperity Index.

## Data Availability

Not applicable.

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
