# Peer review of "Persistence of Schistosomiasis-Related Morbidity in Northeast Brazil: An Integrated Spatio-Temporal Analysis"

_tropicalmed, 2021, doi:10.3390/tropicalmed6040193_

Round 1
Reviewer 1 Report
I would like to suggest that authors could analyze schistosomiasis data with environmental factors that are related to schistosomiasis distribution, e.g. land surface temperature, altitude, rainfall, water temperature, and etc. The results can show the cluster of this disease distribution correlated with environmental factors and can be used for focused interventions.
Author Response
Response to Reviewer 1 Comments
Point 1: I would like to suggest that authors could analyze schistosomiasis data with environmental factors that are related to schistosomiasis distribution, e.g. land surface temperature, altitude, rainfall, water temperature, and etc. The results can show the cluster of this disease distribution correlated with environmental factors and can be used for focused interventions.
Response 1: We agree with the questions regarding the influence of environmental factors, especially the availability of waterways that are potential breeding sites for the intermediate host. However, the objective of the present study was to bring into perspective an epidemiological analysis of cases of the disease from different sources of data, from epidemiological surveillance and hospital admissions. The significant recent increase of poverty and inequality in the country, coupled with reduced control actions reinforce the importance of this study. Issues regarding social and environmental determinants of schistosomiasis have been revised throughout the text for clarity and accuracy.
Reviewer 2 Report
Schistosomiasis is a very important helminth NTD with enormous economic impacts. The topic is very important for policy makers, epidemiologist and health workers of the country and other endemic territories as well. The manuscript contains the saga of battle against schistosomiasis. However, before accepting the manuscript for publication in the journal the following issues must be addressed…
Introduction: Give more emphasis on the impact of the disease, e.g., disease burdens, mortality, socio-economic impacts etc.
Materials and Methods: Describe all attempts under several subheadings. It is a coprological study but surprisingly the method employed the study had not been mentioned in this section. Briefly describe the coprological technique. The authors made a talkative description of the hospital set up and statistical analysis, which the major part of the M&M section; please make a very short account. Method of the survey can be presented as a questioner as a supplemental file.
Results: Tables are very long; make them short and reader friendly; some data can be presented in the supplemental file.
Discussion: At the beginning, add few sentences to introduce your works and key findings.
Conclusions: Make clear cut conclusions on the basis of the results. Rigorous editing is essential.
Ref: Add the recent studies on schistosomiasis e.g., Anisuzzaman and Tsuji 2020, Parasitol Int.; Frahm et al 2019, PLoS NTD; Anisuzzaman et al 2021 Front Immunol etc. among others.
Author Response
Response to Reviewer 2 Comments
Point 1: Schistosomiasis is a very important helminth NTD with enormous economic impacts. The topic is very important for policy makers, epidemiologist and health workers of the country and other endemic territories as well. The manuscript contains the saga of battle against schistosomiasis. However, before accepting the manuscript for publication in the journal the following issues must be addressed.
Response 1: Thank you for your comments.
Point 2: Introduction: Give more emphasis on the impact of the disease, e.g., disease burdens, mortality, socio-economic impacts etc.
Response 2:
There is a dynamic cycle in which the disease acts as both cause and consequence of poverty, contributing to worsening existing socioeconomic conditions. The potential for schistosomiasis to cause disability and death in affected persons is high, although official records are limited, as is research on the burden of disease in endemic countries. According to the Global Burden of Disease Study 2016, the global burden of schistosomiasis is estimated at 1.9 million disability-adjusted life years (DALYs) [18]. In Brazil, schistosomiasis represents the second leading NTD among those analyzed, after Chagas disease [[Martins-Melo FR, Carneiro M, Ramos AN Jr, Heukelbach J, Ribeiro ALP, Werneck GL. The burden of Neglected Tropical Diseases in Brazil, 1990-2016: A subnational analysis from the Global Burden of Disease Study 2016. PLoS Negl Trop Dis. 2018 Jun 4;12(6):e0006559. doi: 10.1371/journal.pntd.0006559. PMID: 29864133; PMCID: PMC6013251]]. In another study, Nascimento and collaborators, in 2018 and 2019, conducted a study in a Northeastern state that showed an average quality of life score of 31.26 QALYs for the population with the chronic digestive forms of schistosomiasis and a total estimated cost per hospitalized case of R$136,087,909.29 (calculation from dollars at the time). It was also estimated at 230,991.75 DALYs, with the majority (219,623 - 95%) represented by the component of Years of Life with Disability [19].
In the amended manuscript, we included the following references:
GBD 2016 DALYs and HALE contributors. Global, regional and national disability-adjusted life years (DALYs) for 333 diseases and injuries and healthy life expectancy (HALE) for 195 countries and territories, 1990-2016: a systematic review for the Global Burden of Disease Study 2016. Lancet 390, 1260-1344 (2017).
- Nascimento GL, Domingues ALC, Ximenes RAA, Itria A, Cruz LN, Oliveira MRF. Quality of life and quality-adjusted life years of chronic schistosomiasis mansoni patients in brazil in 2015. transactions of the royal society of tropical medicine and hygiene, v. 112, p. 238-244, 2018.
- Nascimento GL, Pegado HM, Domingues ALC, Ximenes RAA, Itria A, Cruz LN, Oliveira MRF. The cost of a disease targeted for elimination in Brazil: the case of schistosomiasis mansoni. Mem. Inst. Oswaldo Cruz; 114: e180347, 2019.
Point 3: Materials and Methods: Describe all attempts under several subheadings. It is a coprological study but surprisingly the method employed the study had not been mentioned in this section. Briefly describe the coprological technique. The authors made a talkative description of the hospital set up and statistical analysis, which the major part of the M&M section; please make a very short account. Method of the survey can be presented as a questioner as a supplemental file.
Response 3: We appreciate your considerations. We carried out a general revision of the item referring to methods for greater textual clarity and precision.
Point 4: Results: Tables are very long; make them short and reader friendly; some data can be presented in the supplemental file.
Response 4:
In fact, in this manuscript we carry out a broad analysis with the Brazilian municipalities (5570) as the unit of analysis, with the possibility of integrating variables to better contextualize morbidity patterns in time and space. Despite the extension, these are relevant data in the analysis. The proposal is to insert table 1 as supplementary material to the article, keeping the other tables in their current format.
Point 5: Discussion: At the beginning, add few sentences to introduce your works and key findings.
Response 5: We have reviewed the discussion carefully in order to qualify the introductory part in terms of the context of the study and the main findings.
Point 6: Conclusions: Make clear cut conclusions on the basis of the results. Rigorous editing is essential.
Response 6: The manuscript conclusions were revised accordingly.
Point 7: Ref: Add the recent studies on schistosomiasis e.g., Anisuzzaman and Tsuji 2020, Parasitol Int.; Frahm et al 2019, PLoS NTD; Anisuzzaman et al 2021 Front Immunol etc. among others.
Response 7: The indicated references were inserted in the manuscript.
Globally, approximately 250 million people develop schistosomiasis, resulting in 1430,000 DALYs (disability-adjusted life years) per year. Despite advances, there are critical limitations in the science resulting in a restriction of candidate schistosomiasis vaccines (Sm28GST/Sh28GST, Sm-p80, Sm14 and Sm-TSP-1/SmTSP-2) reaching the different phases of clinical trials. Thus, increasing efforts are needed to achieve the WHO targets set for NTD control [51].
Reviewer 3 Report
At a broad level, this paper includes some potentially interesting findings and trends. However, while the authors state in the Discussion that operational aspects and irregularities in surveillance testing may have influenced findings, overall there is insufficient context provided about the intent and intensity of data capture, the inherent biases of the data sources available, and the impacts of changes to data collection and/or reporting mechanisms over time. I wonder, for example, whether part of the public health response is to intensify surveillance in areas identified as having highest prevalence, and/or whether there are variable triggers for schistosomiasis testing or reporting of results in areas known to have greater disease burden. These factors could significantly influence the likelihood of diagnosing schistosomiasis or capturing it as a principal or secondary diagnosis for those admitted to hospital. Understanding whether or not there is uniformity in surveillance mechanisms (population tested, frequency etc), laboratory testing protocols, treatment programs (eg mass praziquantel therapy), reporting requirements or other factors is crucially important of interpreting the results.
A second overall comment related to the novelty of the findings. A quick literature search reveals some similar and potentially relevant papers, including: Ref 30 (Paz et al); Paz et al, DOI: 10.1016/j.actatropica.2021.105897; Ref 7 (Silva et al). The authors should identify the novel messages from their work.
Minor comments:
It might be helpful for the authors to specifically mention the impact of the time lag between control of disease transmission and admissions/deaths related to chronic complications; even if transmission was able to be completely interrupted today, ongoing morbidity among already-infected individuals is to be expected. Additionally, a brief mention of the rollout of mass treatment over time would be useful.
Line 49: S mansoni is also highly prevalent in Africa, so saying that it has high morbidity “particularly in Brazil” seems misleading, unless the authors are meaning to specifically be referring to the burden in South America.
Line 92: What does “access to sewage collection” mean?
Methods, lines 108-113: The authors should be clearer about what relevant data are collected in SISPCE and SINAN, and what reporting obligations clinicians and others have relevant to schistosomiasis cases. From what is written, it seems that SISPCE collects data on surveillance activities of faecal surveys – how often are these performed and who is involved in providing samples? Why do the number of tests performed vary so much over the years? As per my overall comment above, if certain high-risk areas were targeted in some years for greater surveillance efforts, this would impact on findings. Also, it is unclear whether SINAN is intended to collect only severe cases or all cases, and what the criteria for schistosomiasis testing are. Results: Lines 201-2: What do the authors mean by “The states of Sergipe and Alagoas presented a differentiated pattern, with higher positivity rates”?
The Results section needs to be reworded so that it is clearer. For example “Not living in the state capital was associated with the highest occurrence of the disease…..” should be “Living outside the state capital….” (or similar). Much of the text literally duplicates what is in the tables, when it would be better to convert the text to make it easy for readers to follow the main messages. Just as an example, using phrases such as (line 238-9) “Municipalities of "small size II….” is unhelpful for readers – instead describe results according to how population size impacted on findings. I would suggest limit the abbreviations used. Significant editing is needed.
Are all the tables needed, given the figures also provide some similar results? In Table 3, why are different time periods used?
Author Response
Response to Reviewer 3 Comments
Point 1: At a broad level, this paper includes some potentially interesting findings and trends. However, while the authors state in the Discussion that operational aspects and irregularities in surveillance testing may have influenced findings, overall there is insufficient context provided about the intent and intensity of data capture, the inherent biases of the data sources available, and the impacts of changes to data collection and/or reporting mechanisms over time. I wonder, for example, whether part of the public health response is to intensify surveillance in areas identified as having highest prevalence, and/or whether there are variable triggers for schistosomiasis testing or reporting of results in areas known to have greater disease burden. These factors could significantly influence the likelihood of diagnosing schistosomiasis or capturing it as a principal or secondary diagnosis for those admitted to hospital. Understanding whether or not there is uniformity in surveillance mechanisms (population tested, frequency etc), laboratory testing protocols, treatment programs (eg mass praziquantel therapy), reporting requirements or other factors is crucially important of interpreting the results.
A second overall comment related to the novelty of the findings. A quick literature search reveals some similar and potentially relevant papers, including: Ref 30 (Paz et al); Paz et al, DOI: 10.1016/j.actatropica.2021.105897; Ref 7 (Silva et al). The authors should identify the novel messages from their work.
Response 1:
1- Our analysis was based on secondary data from different health information systems, reflecting the actions of epidemiological surveillance and the hospital health care network. Practical experience in the country with the schistosomiasis surveillance system reveals that the execution of the control policy and the monitoring of its guidelines is extremely heterogeneous among the different municipalities and states in the endemic region and we agree that this aspect should be integrated into the text. In this sense, we carried out extensive textual revision to integrate these aspects into the manuscript.
2- We performed the suggested adjustments including more recent studies on schistosomiasis, as suggested.
Point 2: Minor comments:
It might be helpful for the authors to specifically mention the impact of the time lag between control of disease transmission and admissions/deaths related to chronic complications; even if transmission was able to be completely interrupted today, ongoing morbidity among already-infected individuals is to be expected. Additionally, a brief mention of the rollout of mass treatment over time would be useful.
Line 49: S mansoni is also highly prevalent in Africa, so saying that it has high morbidity “particularly in Brazil” seems misleading, unless the authors are meaning to specifically be referring to the burden in South America.
Response 1: We carry out textual review to clarify the issues indicated by the reviewer.
During the period of this study, the Brazilian MoH recommended the development of control actions with positivity rates above 50%. Despite the current recommendations, the states of the Northeast Region had technical-operational limitations for development, except in the state of Pernambuco, which established in 2011 a state policy to prioritize the control of NTDs - called the SANAR Program (Integrated Plan of Actions for Confronting Neglected Diseases), in which schistosomiasis, among eight other NTDs, was defined as a priority. A study conducted by Facchini and collaborators in 2018 evaluated that the political decision made by the state of Pernambuco to implement the SANAR Program in 2011 impacted the reduction of the schistosomiasis burden. This program was effective in reducing the occurrence of the disease in hyperendemic areas in that state, with a more consistent operational response in areas with two cycles of collective treatment [49].
We have included the following reference:
Luiz Augusto Facchini et al. Assessment of a Brazilian public policy intervention to address schistosomiasis in Pernambuco state: the SANAR program,2011-2014. BMC Public Health (2018) 18:1200.
Line 92: What does “access to sewage collection” mean?
Response 2: Households that have access to garbage collection and sewage treatment. We have adjusted the text for clarification.
Point 3: Methods, lines 108-113: The authors should be clearer about what relevant data are collected in SISPCE and SINAN, and what reporting obligations clinicians and others have relevant to schistosomiasis cases. From what is written, it seems that SISPCE collects data on surveillance activities of faecal surveys – how often are these performed and who is involved in providing samples? Why do the number of tests performed vary so much over the years? As per my overall comment above, if certain high-risk areas were targeted in some years for greater surveillance efforts, this would impact on findings. Also, it is unclear whether SINAN is intended to collect only severe cases or all cases, and what the criteria for schistosomiasis testing are. Results: Lines 201-2: What do the authors mean by “The states of Sergipe and Alagoas presented a differentiated pattern, with higher positivity rates”?
Response 3: According to the recommendations of the Ministry of Health for endemic areas of schistosomiasis, it is recommended to register in SINAN only severe cases and/or cases arising from outbreaks of the disease. We performed adjustments throughout the manuscript to further clarify these review points.
Point 4: The Results section needs to be reworded so that it is clearer. For example “Not living in the state capital was associated with the highest occurrence of the disease…..” should be “Living outside the state capital….” (or similar). Much of the text literally duplicates what is in the tables, when it would be better to convert the text to make it easy for readers to follow the main messages. Just as an example, using phrases such as (line 238-9) “Municipalities of "small size II….” is unhelpful for readers – instead describe results according to how population size impacted on findings. I would suggest limit the abbreviations used. Significant editing is needed.
Are all the tables needed, given the figures also provide some similar results? In Table 3, why are different time periods used?
Response 4: The tables presented in the study are relevant due to the different perspectives brought. The analysis over three years also helps to assess the trend in the distribution of the disease in space and over time, making it possible to infer possible interventions developed in the territories during this period of study.
Reviewer 4 Report
This manuscript focuses on the application of geospatial tools and statistical modelling to describe the dynamics of intestinal schistosomiasis across the northeastern states of Brazil. The effort to collate (and curate) data from public repositories, spanning from 2001 to 2017, is remarkable but, despite these analyses show temporal trends in the number of cases and hospital admissions, geospatial and statistical analyses do not seem fully convincing. Nevertheless, this manuscript furthers previous work by Silva et al. 2019 (https://doi.org/10.1590/0037-8682-0458-2018) and highlight regions where public health interventions may be urgently needed. A number of major and minor concerns are listed below.
MAJOR COMMENTS:
Lines 46-51. Schiostosomiasis is not only associated with absence of sanitation and hygiene, this section could be developed a bit further and take into account the presence of freshwater bodies acting as transmission sites, together with intermediate host distribution and regional problematics that hinder disease control measures. Also briefly develop the clinical forms and pathology that schistosomiasis causes and why it may lead to death.
Materials & Methods. This section is extremely long-winded and would benefit from being split into subsections. The writing is informal at times. Do the classifications made (lines 135-144) all come from reference no. 15? If not, how were poverty and aridity established? Furthermore, what is the authors’ definition of aridity since, to my knowledge, rainfall and schistosomiasis prevalence are not connected. Do the authors mean freshwater coverage of the territory? This should be further explained and the reason why it was included in the analysis should be stated.
Lines 150-167. The application of chi-square was done including how many variables? What is the author’s justification for using this specific test with so many different variables? In my opinion, this is a perfect recipe for spurious significance and the authors should reconsider re-doing the statistical analysis with a more appropriate test (e.g. generalized linear models?). Similarly, I have never come across the Poisson Joinpoint; what is the authors’ rationale for using this, as well as Monte Carlo permutations? What do they aim to prove by using these tests and how do they justify their application?
Lines 168-181. There is no reference to the source of SMA and SMR. Furthermore, the long-winded sections hides the fact that there seems to be no geospatial analysis but only tables and numbers translates into map visualization. The writing should be simplified (e.g. the Jenks classification is only a methodology for binning, but how the authors binned the data does not seem to be clearly mentioned) and, if spatial autocorrelation was applied, its methodology should be highlighted since it does not seem to be explained.
Discussion. Much like the rest of the manuscript, the Discussion is long-winded and a succession of paragraphs that fail to clearly highlight the findings of the study. This section should be reduced accordingly with the revisions listed herein; it should provide clear discussion of the results, a clear picture of the current situation in northeastern Brazil and how public health policies could be implemented to control schistosomiasis.
Lines 380-386. Life outside the state capital does not mean a rural life and this association (i.e. rural = outside the state capital) should not be made since it does not take into account the many urban and suburban areas of a state. The application of urbanization indices is much more appropriate than splitting the data into inside/outside the state capital.
Lines 387-395. As mentioned above, what is the authors’ definition of aridity and where has this variable originated from? If it refers to freshwater coverage of a territory OK but it should not be part of any analysis if it refers to rainfall.
Supplementary Materials. Should the curated database be made available as supplementary material in agreement with current trends about data transparency and open source?
MINOR COMMENTS:
Lines 32-42. Does the Abstract need splitting into objective, methods, results, etc? If not remove them and modify the text accordingly.
Line 46. Schistosoma and Schistosoma mansoni always in italics not just here but throughout the whole manuscript.
Line 49. Remove ‘.
Line 55. Reference no. 4 is not appropriate, for the latest guidelines by the WHO see Casulli 2021 (https://doi.org/10.1371/journal.pntd.0009373).
Lines 64-65. See also Zoni et al. 2016 (https://doi.org/10.1371/journal.pntd.0004493), it may be an important reference to include.
Line 69-73. Sergipe and Bahia. A bit more details could be added regarding transmission foci across the northeastern region of Brazil.
Lines 82-84. Performed. Sergipe and Bahia.
Lines 87-90. Vulnerability is a recurring theme throughout this manuscript (see also Line 71) but it is never properly described and this section is a good opportunity to define it.
Lines 94-96. This sentence does not read well.
Lines 103-107. Include the definition of secondary data and, if possible, any references or websites for the different surveillance programmes that have been mentioned.
Lines 115-118. The reference regarding diagnostics guidelines by the WHO may be missing.
Lines 128-132. What is B65? A reference seems to be missing for this paragraph.
Lines 135-144. This section could be replaced by a table listing variable name, source, etc.
Lines 164-167. This sentence may belong to the Results sections.
Lines 188-194. Define the level of aggregation although it would be helpful to state it earlier in the methodology as well.
Results. This section should also be split into subsections, as it stands it is hard to follow.
Line 196. Were performed.
Figure 1. Change from Portuguese to English and, in the legend, state that the area subject of the study is highlighted in grey.
Table 1. In the text, the authors mentioned grouping into triennia and the same could be done here to reduce the sizing of the table; furthermore, instead use “Total” instead of “Northeast regions” as done for the following tables.
Figure 2. This figure is way too small to understand and each acronym and panel should be explained in the legend.
Tables 2 and 3. P-value and not p-valor; these tables are massive, could they belong to a supplementary material rather than the main text?
Lines 324-326. Any reference and/or evidence for this reduction?
Line 331. Morbimortality is not correct, it is also present elsewhere in the manuscript and this terminology should be amended.
Line 391. Biomphalaria in italics.
Lines 423-432. This sentences are way too long and the language is too informal. Parameters rather than tools?
Lines 439-443. This paragraph does not fit the overall discussion and should be removed.
Lines 448-449. This sentence is not clear.
Author Response
Response to Reviewer 4 Comments
Point 1: This manuscript focuses on the application of geospatial tools and statistical modelling to describe the dynamics of intestinal schistosomiasis across the northeastern states of Brazil. The effort to collate (and curate) data from public repositories, spanning from 2001 to 2017, is remarkable but, despite these analyses show temporal trends in the number of cases and hospital admissions, geospatial and statistical analyses do not seem fully convincing. Nevertheless, this manuscript furthers previous work by Silva et al. 2019 (https://doi.org/10.1590/0037-8682-0458-2018) and highlight regions where public health interventions may be urgently needed. A number of major and minor concerns are listed below.
Response 1: Thank you for your comments.
MAJOR COMMENTS
Point 2: Lines 46-51. Schistosomiasis is not only associated with absence of sanitation and hygiene, this section could be developed a bit further and take into account the presence of freshwater bodies acting as transmission sites, together with intermediate host distribution and regional problematics that hinder disease control measures. Also briefly develop the clinical forms and pathology that schistosomiasis causes and why it may lead to death.
Response 2: Thank you for your comments. In fact, although schistosomiasis has classically rural characteristics, where people have contact with water collections with the presence of intermediate hosts ("snails") releasing cercariae - the infective form of the parasite, there is a process of migration and urbanization that affects populations in large cities, particularly in people living in areas with poor basic sanitation.
Point 3: Materials & Methods. This section is extremely long-winded and would benefit from being split into subsections. The writing is informal at times. Do the classifications made (lines 135-144) all come from reference no. 15? If not, how were poverty and aridity established? Furthermore, what is the authors’ definition of aridity since, to my knowledge, rainfall and schistosomiasis prevalence are not connected. Do the authors mean freshwater coverage of the territory? This should be further explained and the reason why it was included in the analysis should be stated.
Response 3: Thank you for your comments. We have carried out careful review of the methods section. In this manuscript, we include information on the existence of schistosomiasis cases in semi-arid municipalities due to the presence of temporary freshwater collections, the presence of intermediate hosts transmitting schistosomiasis and also the fact that some municipalities are part of the São River Project, transposition of the Francisco River, where there will be the possibility of greater access to water in the region and, consequently, a potential risk of establishing or expanding transmission foci in these areas, as evidenced by the study by Silva Filho et al (2017).
Point 4: Lines 150-167. The application of chi-square was done including how many variables? What is the author’s justification for using this specific test with so many different variables? In my opinion, this is a perfect recipe for spurious significance and the authors should reconsider re-doing the statistical analysis with a more appropriate test (e.g. generalized linear models?). Similarly, I have never come across the Poisson Joinpoint; what is the authors’ rationale for using this, as well as Monte Carlo permutations? What do they aim to prove by using these tests and how do they justify their application?
Response 4: The statistical analysis included are based on other studies that worked on NTDs, as well as ecological studies. In the following we present some studies that performed similar analysis techniques (https://doi.org/10.1017/S0031182016002341, http://dx.doi.org/10.2471/BLT.15.152363, https://doi.org/10.1093/trstmh/trv069). Joinpoint time series analysis has been used in several studies to evaluate the variation over time of mortality and incidence trends. In general, this technique allows the evaluation of whether health policies are being effective, allowing the control of the disease, for example, in the case of schistosomiasis (https://doi.org/10.1016/j.parepi.2016.03.002, https://doi.org/10.1016/j.actatropica.2021.105948)
Point 5: Lines 168-181. There is no reference to the source of SMA and SMR. Furthermore, the long-winded sections hides the fact that there seems to be no geospatial analysis but only tables and numbers translates into map visualization. The writing should be simplified (e.g. the Jenks classification is only a methodology for binning, but how the authors binned the data does not seem to be clearly mentioned) and, if spatial autocorrelation was applied, its methodology should be highlighted since it does not seem to be explained.
Response 5: We adjusted the text for clarity and precision. We included references to technical analysis procedures via SMA and SMR (Haining RP. The moving average model for spatial interaction. Trans Inst Br Geogr. 1978; 3(2):202; Bernardinelli L, Montomoli C. Bayes empirical versus fully analysis Bayesian geographic data variation in disease risk Stat Med. 1992; 11 (June 1990): 983–1007.). These techniques allow to present spatial patterns based on special descriptive data; emphasizing they do not refer to geospatial analysis. For the Jenks classification, we describe that this technique was used to define the categories of the analyzed indicators. We did not use autocorrelation analysis in this study, and descriptive analyzes significantly present the proposed problem.
Point 6: Discussion. Much like the rest of the manuscript, the Discussion is long-winded and a succession of paragraphs that fail to clearly highlight the findings of the study. This section should be reduced accordingly with the revisions listed herein; it should provide clear discussion of the results, a clear picture of the current situation in northeastern Brazil and how public health policies could be implemented to control schistosomiasis.
Response 6: We have revised the discussion in the amended manuscript.
Point 7: Lines 380-386. Life outside the state capital does not mean a rural life and this association (i.e. rural = outside the state capital) should not be made since it does not take into account the many urban and suburban areas of a state. The application of urbanization indices is much more appropriate than splitting the data into inside/outside the state capital.
Response 7: We performed a textual review to include the issues raised.
Point 8: Lines 387-395. As mentioned above, what is the authors’ definition of aridity and where has this variable originated from? If it refers to freshwater coverage of a territory OK but it should not be part of any analysis if it refers to rainfall.
Response 8: Thank you for your observations. We can consider the above answer in point 3.
Point 9: Supplementary Materials. Should the curated database be made available as supplementary material in agreement with current trends about data transparency and open source?
Response 9: Answer 9: Yes, the study data are publicly accessible and we have reviewed the necessary links for easy access.
MINOR COMMENTS:
Point 10: Lines 32-42. Does the Abstract need splitting into objective, methods, results, etc? If not remove them and modify the text accordingly.
Response 10: We follow the instructions for authors provided by the journal.
Point 11: Line 46. Schistosoma and Schistosoma mansoni always in italics not just here but throughout the whole manuscript.
Point 12: Line 49. Remove ‘.
Response 12: Adjusted.
Point 13: Line 55. Reference no. 4 is not appropriate, for the latest guidelines by the WHO see Casulli 2021 (https://doi.org/10.1371/journal.pntd.0009373).
Response 13: Thank you for the comments - we have adjusted this reference.
Point 14: Lines 64-65. See also Zoni et al. 2016 (https://doi.org/10.1371/journal.pntd.0004493), it may be an important reference to include.
Response 14: We have included this reference.
Point 15: Line 69-73. Sergipe and Bahia. A bit more details could be added regarding transmission foci across the northeastern region of Brazil.
Response 15: Yes, we consider our results as well as Silva and collaborators (2019) when doing the analyses of space-time trends in these two areas.
Point 16: Lines 82-84. Performed. Sergipe and Bahia.
Response 16: We changed the term that was inconsistent in writing. And adjusted the "e" to "and".
Point 17: Lines 87-90. Vulnerability is a recurring theme throughout this manuscript (see also Line 71) but it is never properly described and this section is a good opportunity to define it.
Response 17: We have included the definition in the manuscript.
Point 18: Lines 94-96. This sentence does not read well.
Response 18: We adjust the text for greater precision and quality.
Point 19: Lines 103-107. Include the definition of secondary data and, if possible, any references or websites for the different surveillance programmes that have been mentioned.
Response 19: We adjust the text for greater precision and quality.
Point 20: Lines 115-118. The reference regarding diagnostics guidelines by the WHO may be missing.
Response 20: We adjust the text for greater precision and quality.
Point 21: Lines 128-132. What is B65? A reference seems to be missing for this paragraph.
Response 21: We adjust the text for greater precision and quality.
Point 22: Lines 135-144. This section could be replaced by a table listing variable name, source, etc.
Response 22: For a better understanding of the reader, we chose to present the variables textually. We tried to follow the instructions for the authors throughout the construction of the entire manuscript.
Point 23: Lines 164-167. This sentence may belong to the Results sections.
Response 23: This sentence presents how the results should be interpreted. It is similar to that quoted in the method of other authors who work with this technique (Martins-Melo FR, 2014: https://www.sciencedirect.com/science/article/pii/S2405673115300854; Ferreira AF, 2020: https://onlinelibrary.wiley.com/doi/full/10.1111/tmi.13343)
Point 24: Lines 188-194. Define the level of aggregation although it would be helpful to state it earlier in the methodology as well.
Response 24: The data were aggregated by county of residence of the schistosomiasis cases.
Point 25: Results. This section should also be split into subsections, as it stands it is hard to follow.
Response 25: We have included appropriate subsections for better organization and presentation of the methods section.
Point 26: Line 196. We have included the subsections in the method.
Response 26: We adjust the text for greater precision and quality.
Point 27: Figure 1. Change from Portuguese to English and, in the legend, state that the area subject of the study is highlighted in grey.
Response 27: We made the indicated adjustments. We changed the image, putting the name of the regions in English. The highlighted study area is represented in dark grey.
Point 28: Table 1. In the text, the authors mentioned grouping into triennia and the same could be done here to reduce the sizing of the table; furthermore, instead use “Total” instead of “Northeast regions” as done for the following tables.
Response 28: Thank you for your observation. We will use this form in future studies to facilitate the reader's understanding. We made the indicated adjustments in terms of the variation of the total in the Northeast region.
Point 29: Figure 2. This figure is way too small to understand and each acronym and panel should be explained in the legend.
Response 29: We made the indicated adjustments. We changed the size of the texts and make up the figure and its dimensions.
We have adjusted the colors of the lines in the graph to make it easier to see.
Point 30: Tables 2 and 3. P-value and not p-valor; these tables are massive, could they belong to a supplementary material rather than the main text?
Response 30: We adjust the text for greater precision and quality. We replace "p-value" by "P-value ".
Point 31: Lines 324-326. Any reference and/or evidence for this reduction?
Response 31: The studies presented in this manuscript show this reduction trend.
Point 32: Line 331. Morbimortality is not correct, it is also present elsewhere in the manuscript and this terminology should be amended.
Response 32: We adjust the text for greater precision and quality.
Point 33: Line 391. Biomphalaria in italics.
Response 33: OK, done.
Point 34: Lines 423-432. This sentences are way too long and the language is too informal. Parameters rather than tools?
Response 34: We prefer using this term - it follows the pattern of the study.
Point 35: Lines 439-443. This paragraph does not fit the overall discussion and should be removed.
Response 35: We consider the paragraph mentioned in the review to be important for further contextualizing the study.
Point 36: Lines 448-449. This sentence is not clear.
Response 36: We adjusted the text for greater precision and quality.
Round 2
Reviewer 3 Report
I can't see any edits to the manuscript recognising my comment regarding the time lag between control of disease transmission and admissions/deaths related to chronic complications, nor any description of mass treatment programs for schistosomiasis.